# The regulatory domains of the lipid exporter ABCA1 form domain swapped latches

Stephen G. Aller[1]*, Jere P. Segrest[2]

**1** Department of Pharmacology and Toxicology, University of Alabama at Birmingham, Birmingham, AL, United States of America, **2** Division of Cardiovascular Medicine, Department of Medicine, Vanderbilt University Medical Center, Nashville, TN, United States of America

\* sgaller@uab.edu

**Data Availability Statement:** The data are available at Protein Data Bank entry, ID 7ROQ.

**Funding:** Funded by NIH grant P01HL128203. The funders had no role in study design, data collection

## Abstract

ABCA1 and ABCA4 are enigmatic because they transport substrates in opposite directions yet share >50% amino acid identity. ABCA4 imports lipid conjugates but ABCA1 exports lipids. Both hydrolyze ATP to drive transport, and both contain cytoplasmic regulatory domains (RDs) following nucleotide-binding domains (NBDs) in the primary structure. The tertiary structures of several ABC importers, including ABCA4, show that each RD forms a domain-swapped latch that locks onto the opposing RD and holds the NBDs close together. Crucially, sequences encoding the RDs and their bridges are among the most conserved in the entire ABC-A subfamily. In the original cryo-EM structure of ABCA1, the RDs were modeled without crossover. After close inspection of that cryo-EM density map and the recent structure of ABCA4, we propose that the RDs of ABCA1 also form a domain-swapped latch. A refined ABCA1 model containing latches exhibited significantly improved overall protein geometry. Critically, the conserved crossover sequence leading to the RD-domain swap is directly supported by the original cryo-EM density map of ABCA1 and appears to have been overlooked. Our refined ABCA1 model suggests the possibility that ABCA1, despite being an exporter, has highly restrained NBDs that suggest a transport mechanism that is distinct from 'alternating access'.

## Introduction

Transporters in the ATP-binding cassette (ABC) superfamily use energy derived from ATP to carry substrates across membrane bilayers [1]. Their common architecture includes transmembrane domains (TMDs) and nucleotide-binding domains (NBDs). ABC half-transporters contain a single TMD and a single NBD that form either a heterodimer or homodimer. ABC full-transporters are a single polypeptide that contains two TMDs and two NBDs [2].

Extensive structure-function studies of the bacterial maltose importer have revealed the presence of two domain-swapped regulatory domains (RDs) that immediately follow the NBDs in the amino acid chain. The RDs cross over the two-fold symmetry axis to form a pair of latches. The latches constrain the NBDs, holding them in close proximity. In no fewer than 11 high-resolution snapshots of the different conformational states in the ATP catalytic cycle,

and analysis, decision to publish, or preparation of the manuscript.

**Competing interests:** The authors have declared that no competing interests exist.

the latches were well-ordered and always in the locked position. The multiple structures included: a nucleotide- and substrate-free resting state [3], substrate-bound but nucleotide-free pre-translocation state [4], a nucleotide- and substrate-bound form [5], four transition-state mimetic structures with substrate [6], and two outward-facing variants [4, 6]. The significance of the latches is highlighted by two early structures of isolated NBD-RDs that were not constrained by transmembrane segments because they were not present. Even under those conditions, the latches were firmly locked in place [7, 8].

The RDs are an impressive feature of the maltose transporter from *E. coli* and, at ~35 kDa together, they are nearly as massive as both NBDs together (~40 kDa). Structures of other prokaryote importers, such as the methionine importer from *E. coli* [9], the LpqY-Sug importer of *Mycobacterium* [10], and the osmoregulatory transporter OpuA [11], also revealed extensive domain-swapping of RDs that form locked latches having masses of 30 kDa, 32 kDa, and 31 kDa, respectively. Much smaller motifs are present in bacterial importers for molybdenum [12, 13], heme [14, 15], amino acids [16], and metal-siderophores [17]. In those cases, however, the motif is a simple extended α-helix that cannot comprise an entire latch-like domain. Yet even in the cases of these minor motifs, the extensions cross over the axis of symmetry and contact the opposing extended α-helix. Taken together, each RD of many bacterial importers cross over the axis of symmetry and provide significant van der Waals and electrostatic contact with the opposing RD. The function appears consistent with the interpretation that the well-defined latches restrict or clamp [18] the extent of NBD separation during ATP cycling that dictates a tweezers-like mechanism of import [7]. Given its prevalence in several bacterial import proteins, the latch structure appears to have arrived very early in the evolution of transporters.

The domain-swapped latch motif is not prevalent in eukaryotic transporters, so it may not be a coincidence that eukaryotic importers are very rare. One well-characterized exception is the ABC full transporter, ABCA4 [19, 20]. This is the only known importer in the ABC-A subfamily but closely related phylogenetically to ABCA1 (**Fig 1**), an ABC full-transporter and well-characterized exporter [21, 22]. In fact, ABCA1 and ABCA4 form one of the most closely related clades in the entire ABC-A subfamily (**S1 Fig**), and the ABCA1 clade appears to be a major evolutionary root of the entire ABC-A subfamily. ABCA4, which transports the retinoid compound from the lumen leaflet to the cytoplasmic leaflet of disk membranes in the eye is found in eukaryotes with complex vision, is lacking in organisms with more primitive eyes such as the horseshoe crab, and is completely absent from invertebrates. ABCA1 is prevalent in both invertebrates and vertebrates.

Five available structures of ABCA4 (3.27–3.40 Å) reveal the presence of a domain-swapped pair of latches (~28 kDa) in this specialized eukaryotic importer [23, 24]. Three structures in the ATP-free state reveal that the greatest gap between the parts of the NBDs that are closest to the membrane, yet the latches in all five structures are tightly locked and the parts of the NBDs closest to the cytoplasm are clamped tight. Amino acid sequences corresponding to the RDs in human ABCA4 and human ABCA1 are 59% identical and ~76% similar, so it is reasonable to expect that the RDs in ABCA1 would also contain domain-swapped latches. However, structural models of the RDs in ABCA1 lacked the crossover, and they suggested that each RD makes contact with the other through only a simple interface. A simple interface for ABCA1, as opposed to a locked latch, might be compelling as it might suggest the possibility that ABCA4 and ABCA1 have different transport mechanisms based on restraints, or lack thereof, imposed on the NBDs during ATP cycling.

Given the high evolutionary conservation of this region of the transporters of ABCA1 and ABCA4, however, we wondered if the 2017 model of the RDs of ABCA1 might be flawed. We therefore constructed an alternative model to see if ABCA1 could, in fact, have locking latches. Below, we rationalize our refined model of ABCA1 containing locked latches, and discuss its implications for transport mechanisms in the ABC-A transporter family.

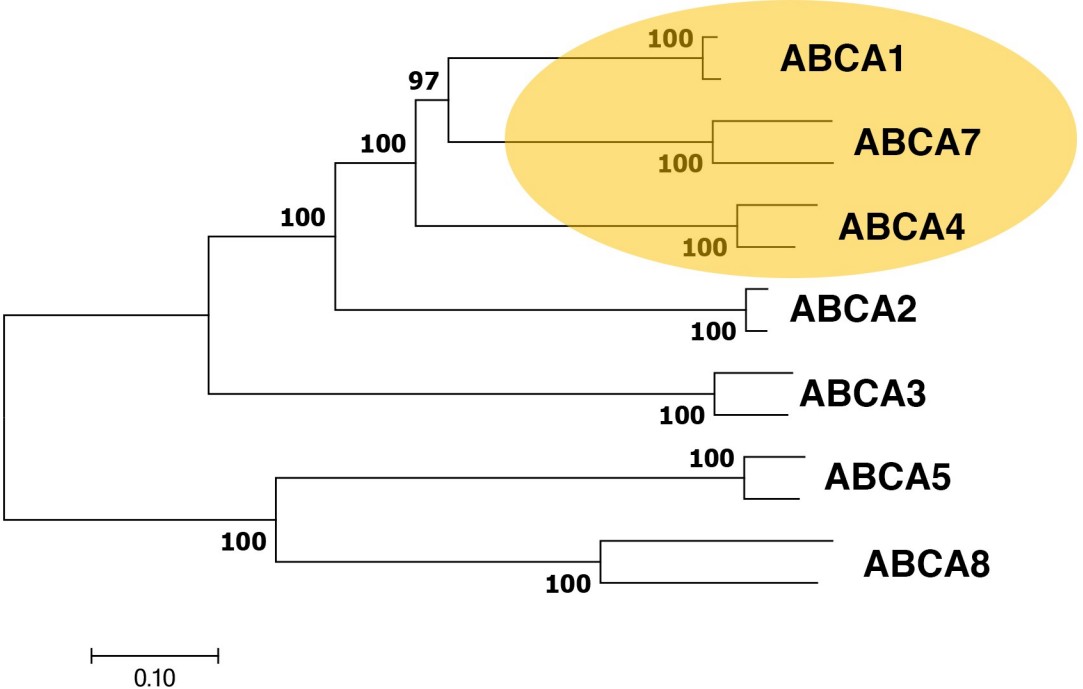

**Fig 1. Phylogenetic relationships between mammalian ABC-A subfamily members.** The optimal tree is drawn to scale using mouse- and human- ABC-A sequences with branch lengths in the same units as those of the evolutionary distances used to infer the tree. Distances are in the units of the number of amino acids substitutions per site. The percentage of replicate trees in which the associated taxa clustered together in the bootstrap test using 500 replicates [37] and are shown at the branchpoints.

## Results

The 2021 structures of ABCA4, which range in resolution from 3.27 Å to 3.40 Å [23, 24], provided high-quality cryo-EM density, technically Coulomb potential density, for two distinct conformations of the transporter, and revealed post-translational modification (glycosylation) and several different ligands (lipids, cholesterol, and the N-ret-PE substrate). The quality of cryo-EM density of the earlier 4.10 Å structure of ABCA1, obtained in 2017 [25], was generally good for the extracellular domains and transmembrane regions, requiring no significant changes in those regions in our refined model.

In contrast, a striking difference between the 2017 model of ABCA1 and the 2021 models of ABCA4 is evident in the polypeptide backbone of the cytoplasmic RDs (**Fig 2**). As with the bacterial importers mentioned above, ABCA4 clearly has domain-swapped RDs that form a pair of latches that hold portions of the NBDs in close proximity.

Close inspection of the cryo-EM density of the 2017 ABCA1 map revealed diminished quality for the second nucleotide binding domain (NBD2) and lower quality for the RDs (**Fig 3**). Contour analysis to reveal the signal-to-noise ratio of the relative cryo-EM density in the complete protein by domains showed that ECD2 > ECD1 > TMD1 ≈ NBD1 > TMD2 ≈ NBD2 > RD1 ≈ RD2. The RDs contained several breaks in the polypeptide chain and the percentage of secondary structure, specifically β-sheet, for the overall protein was significantly lower compared to several other ABC transporter structures (**S1 Table**). Therefore, we were prompted to consider that modeling errors were present in the 2017 structure of ABCA1.

Given the implications for the fundamental transport mechanisms within the ABC-A subfamily, and to shed some light on the discrepancy between the ABCA1 and ABCA4 models, we performed a phylogenetic analysis. Multi-sequence alignment of the RDs showed

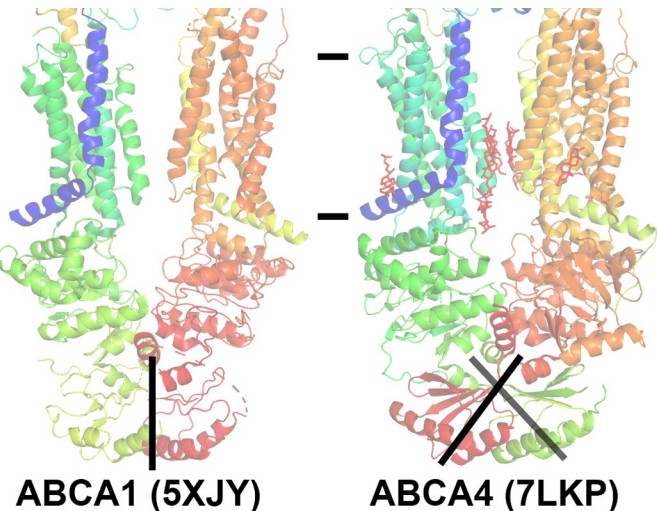

**Fig 2. Difference in modeling the regulator domains (RDs) of ABCA1 and ABCA4.** The published structures of ABCA1 [25] and ABCA4 [23] are shown including PDB IDs. Models are colored from N-terminus to C-terminus as a rainbow gradient from blue to red. The approximate position of the membrane is delineated. RDs of ABCA1 are modeled as a simple contacting interface without crossover, whereas ABCA4 RDs were modeled with crossover. The ABCA4 RDs cross over the axis of symmetry and formed a domain swapped pair of latches to hold a portion of the nucleotide binding domains in close proximity.

remarkable conservation of fundamental structure-function of the ABC-A subfamily, including RD crossover and bridging sequences (**Fig 4**). Several amino acid residues in that region are completely invariant (**Fig 4, panel B**). A larger alignment that included invertebrate ABC-As showed that the sequences involved in cross over and flanking (bridging) were as conserved as those involved in nucleotide binding and the binding of ligands to the protein at the inner leaflet of the plasma membrane (**S2 Fig**).

Given the extensive conservation of sequences corresponding to the domain swapped RD latch in ABCA4 specifically, we were motivated to build and refine a model of ABCA1 that includes domain-swapped latches defined by the RDs. We refined our initial model against the original deposited experimental cryo-EM density [25], and the final model exhibited significantly improved geometric benchmarks, such as 93.4% Ramachandran sidechain favorability

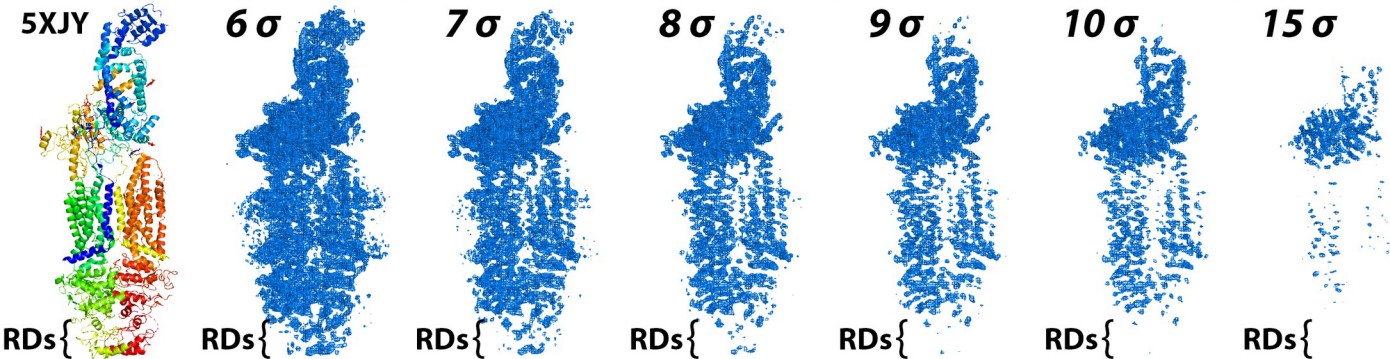

**Fig 3. Contour analysis of the published 2017 cryo-EM structure of ABCA1 and associated experimental cryo-EM density.** The left panel displays the ABCA1 structure [25], PDB ID 5XJY, colored from N-terminus to C-terminus as a rainbow gradient from blue to red. The other panels show the original experimental cryo-EM density (emd-6724) contoured at the displayed sigma contour levels. The position of the cytoplasmic Regulatory Domains (RDs) are indicated with brackets in each panel.

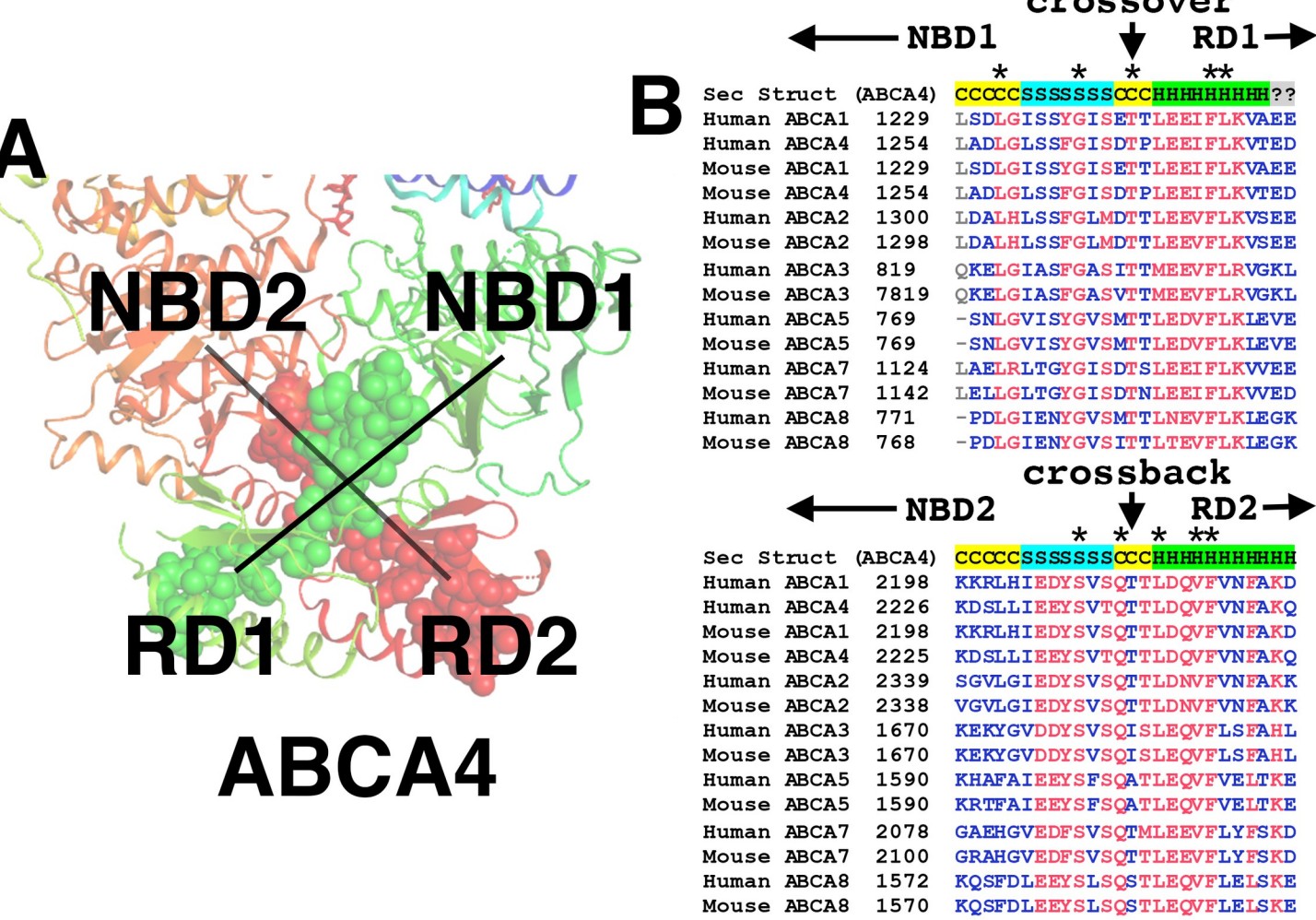

**Fig 4. Conservation of the crossover and flanking sequences in the ABC-A subfamily.** Panel A shows the two crossover motifs as modeled in the ABCA4. The first crossover with flanking sequences are shown as green spheres and connects NBD1 with RD1 by crossing over the pseudo two-fold axis of symmetry. The second motif, which we call the "crossback" is symmetrically related to the first, and connects NBD2 with RD2 (shown as red spheres). Panel B shows the alignment of ABC-A crossover and crossback sequences for mouse- and human- transporters. The secondary structure from ABCA4 is overlaid at the top of each alignment with C = coil, S = β-sheet and H = α-helix. The amino acid residue that resides at the exact axis of symmetry for the first crossover is an invariantly conserved Threonine (human ABCA1 Thr1242). A highly conserved symmetrically related Threonine is present at the axis of symmetry for the crossback (Human ABCA1 Thr2211).

compared to 85.8% favorability in the previous model (**Table 1**). Critically, the short amino acid stretch that spans the pseudo-symmetric axis of ABCA4, which is highly conserved in the subfamily, is a strong feature in the original cryo-EM density map of ABCA1 (**Fig 5**). Taken together, we propose an alternative model of ABCA1 that contains a highly conserved domain-swapped latch that suggests a subtle but distinctly different mechanism between sub-strate import and export for the ABC-A subfamily.

## Discussion

The transport cycle of ATP-binding cassette (ABC) exporters is thought to be governed by rocking motions of the transmembrane regions as part of a mechanism called "alternating access." This mechanism was proposed by Jardetzky almost 60 years ago [26]. Dozens of exporter structures have confirmed significant conformational differences between the

**Table 1. Refinement and model statistics for structures of ABCA1 and ABCA4.**

| | Human ABCA1 | Human ABCA1 (refined) | Human ABCA4 |
|---|---|---|---|
| | PDB 5XJY | PDB 7ROQ | PDB 7LKP |
| CC (mask) | 0.76 | 0.71 | 0.74 |
| CC (volume) | 0.74 | 0.69 | 0.74 |
| CC (peaks) | 0.52 | 0.47 | 0.53 |
| Mean CC for ligands | 0.70 | 0.62 | 0.63 |
| Clashscore, all atoms[a] | 9.20 | 11.8 | 19.9 |
| Rotamer outliers[a] | 1.27% | 0% | 0.60% |
| Ramachandran outliers[a] | 0.16% | 0.00% | 0.05% |
| Ramachandran favored[a] | 85.8% | 93.6% | 94.3% |
| Rama-Z, whole (RMSD) | -6.01 (0.13) | -0.58 (0.20) | -0.89 (0.19) |
| Rama-Z, helix (RMSD) | -3.43 (0.11) | 1.30 (0.18) | 0.84 (0.16) |
| Rama-Z, sheet (RMSD) | -5.20 (0.28) | -1.24 (0.47) | -1.90 (0.39) |
| $C_\beta$ deviations[a] | 0% | 0% | 0% |
| MolProbity score[a,b] | 2.21 | 2.07 | 2.18 |

[a]http://molprobity.biochem.duke.edu (using electron-cloud x-H bond-lengths and no N/Q/H flips).

[b]MolProbity score combines the clashscore, rotamer, and Ramachandran evaluations into a single value.

inward- and outward-facing states implied by the alternating access hypothesis. Structures of inward-facing conformations that are substrate- or nucleotide- free often exhibit a large separation of the NBDs that opens up the transporter to the inner leaflet of the membrane so substrates can enter. This conformation is stabilized by a domain swap that crosses over at TM4/5 and crosses back at TM 10/11. This particular crossover motif is conserved in the ABC-B, ABC-C, and ABC-D subfamilies, and it appears very early in evolution as exemplified by the metal resistance transporter of yeast [27] and by homo-dimeric bacterial lipid exporter floppases such as Sav1866 [28]. Transporters with this motif were recently reclassified into a 'Type IV' system. The ABC-A and ABC-G subfamilies were reclassified as 'Type V' because the first-in-class structures of ABCA1 [29] and ABCG5/ABCG8 [30] lacked the TM crossover.

The Type V structures of ABCA1 and ABCG5/G8 seemed to contain only a simple RD interface without domain swapping yet the NBDs are in close proximity in ATP-free conformations. We questioned this conclusion by closely examining ABCA1, which has significant (~28 kDa) regulatory domains (RDs) at the cytoplasmic faces of the NBDs. ABCA1 RDs were modeled without crossover in the original report, yet they formed a simple contacting interface even when the structure was ATP-free [25]. Interestingly, the more recent higher-resolution structures of ABCA4 [23, 24] clearly show the RDs crossing over to form domain-swapped latches that clamp the NBDs. The latches are locked in all five structures of ABCA4, even though three of the structures are ATP-free.

Although the absence of a domain-swapped latch in ABCA1 would be intriguing, our close examination of phylogeny and the 2017 cryo-EM density map prompted us to dispute the original modeling of the RDs in ABCA1. We now propose that ABCA1, and most likely other members of the ABC-A subfamily, contain the domain-swapped latch. The finding offers a convincing explanation for why the cytoplasmic faces of the nucleotide-binding domains (NBDs) stay in close contact even in the absence of ATP. Our new model of ABCA1 also agrees better with multiple higher-quality structures of ABCA4, which exhibits at least 66% overall amino acid similarity to ABCA1.

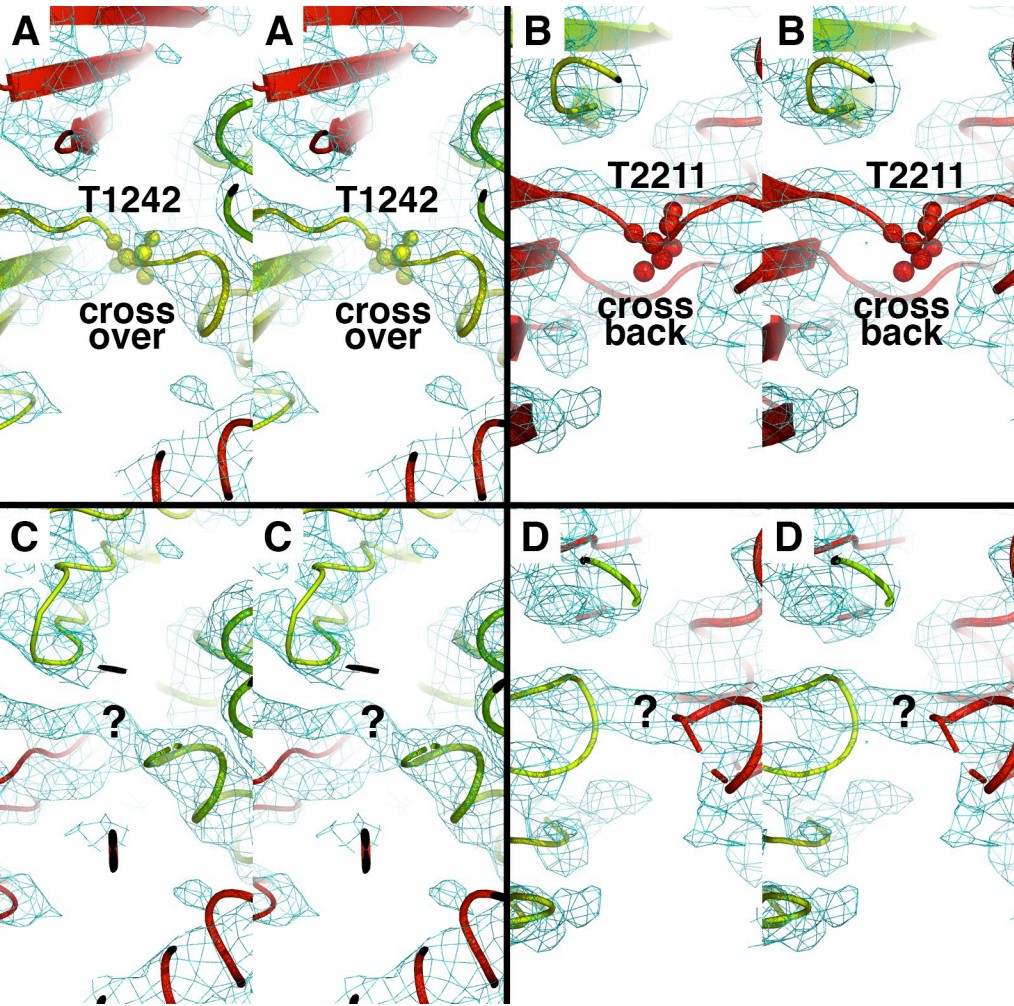

**Fig 5. Original experimental cryo-EM density supports crossover domain swap for ABCA1 regulatory domains.** The four panels are in wall-eyed stereo view. Panels A and B show our new proposed refined ABCA1 model. Panels C and D show the original published model (PDB 5XJY). Two Threonine residues located on the precise axis of pseudo-symmetry at the crossover (panel A, labeled T1242) and crossback (panel B, labeled T2211) are highly conserved amongst all members of the ABC-A subfamily (see Fig 4).

Careful analysis of the conserved ligand-binding residues in the ABC-A subfamily (S2 Fig) and of the cryo-EM density in the original ABCA1 map (S3 Fig) revealed another interesting feature: the steroid moiety of at least two cholesterol (or CHS) ligands are bound between TM helices 4, 5, and 7 of ABCA1 within the inner membrane leaflet. The positions of these two ligands agrees surprisingly well with the relative positions of two corresponding ligands in the ATP-free structure of ABCA4. Cholesterol is not a known substrate of ABCA4, but since the ligand-binding residues in this region are highly conserved in the entire subfamily, it is possible that cholesterol binding at these sites affords structural stability to the transporters. Cholesterol, however, is a likely substrate of ABCA1 [31], and our identification of these two ligands in the original cryo-EM density map raises the tantalizing possibility that one or more of the molecules are poised to be flopped onto the outer leaflet and then translocated to the extracellular region for loading onto HDL (Segrest et al., submitted).

We conclude: 1) ABCA1 contains domain-swapped regulatory domains that appear to function like latches, and 2) the domain swapping is conserved in the entire ABC-A subfamily.

We speculate that the common domain-swapped architecture may imply a mechanism of transport for ABC-A that is quite distinct from the canonical alternating access model [26] shown for other subfamilies. The concept is consistent with the recent reclassification of ABC-A family members as Type V transporters. Type IV transporters (ABC-B, ABC-C and ABC-D) are completely distinct because they lack any RDs and their crossover motif in the TMDs may primarily allow a relatively large separation of NBDs during ATP cycling and alternating access.

## Methods

### Molecular phylogeny

To retrieve putative ABC-A coding sequences with good representation from vertebrates and invertebrates, we searched for the sequence in the full nonredundant dataset in the NCBI BlastP Server. To find potential ancestral invertebrate ABC-A sequences, we performed the search again by using the sequence of human ABCA1 and excluding the Vertebrata taxid (7742). Global alignments were performed using the Clustal method [32] within the Lasergene software package (DNASTAR, Inc.) as previously described [33]. Similar results were obtained with Clustal Omega [34] in the EMBOSS package [35] and standard parameters [36]. Alignments for phylogenetic analyses, including the production of the phylogenetic tree in Figs 1 and S1, were conducted with MEGA7 [37].

### Constructing and refining the alternative ABCA1 model

Close inspection of the atomic model and cryo-EM density were performed using PyMOL [38] and Coot [39] using the published 2017 structure of ABCA1, PDB entry 5XJY [25], and the 2021 structure of ATP-free ABCA4 [23], PDB code 7LKP, and their corresponding cryo-EM density maps (emd_6724 and emd_23409, respectively) obtained from the PDB. Inspection of ABCA1 and original cryo-EM density revealed that no changes were needed prior to refinement for any of the extracellular- or transmembrane regions of the protein. The primary amino acid sequences of the nucleotide binding domains (NBDs) and regulatory domains (RDs) of ABCA4 (resi 916–1277 and 1911–2253) were used to identify the corresponding sequences in ABCA1 using NCBI Cobalt [40]. Disordered sequences not present in the ABCA4 cryo-EM structure were removed prior to alignment and insertion sequences present in either protein but not the other were also removed. A few small segments of the original ABCA1 model in the NBDs that were not well supported by cryo-EM density were also removed. Mutagenesis of the ABCA4-NBD/RDs to ABCA1 sequences was performed with scripts written in perl and PyMOL version 2.0.6 [33]. Refinements of the ABCA1 starting model were performed using RealSpace refinement in phenix [41] at a cutoff of 4.13 Å resolution. After each round, the refined model was inspected with the original cryo-EM density and residues having poor coverage using Density Fit Analysis with Coot [42].

## Supporting information

**S1 Fig. Phylogenetic relationships of the clades formed by the ABC-A subfamily.** Sequences were obtained from the NCBI database according to the methods in the main text. The optimal tree is drawn to scale using the indicated ABC-A sequences with branch lengths in the same units as those of the evolutionary distances used to infer the tree. Distances are in the units of the number of amino acids substitutions per site. The percentage of replicate trees in which the associated taxa clustered together in the bootstrap test using 500 replicates (ref 34) and are shown at the branchpoints. The tree shows that ABCA4 and ABCA7 are closely related to

ABCA1 sequences.
(DOCX)

**S2 Fig. Conservation of amino acid residues involved in fundamental transport functions for the ABC-A subfamily, including invertebrates.** The figure shows a compressed view of the entire alignment for all amino acid residues comprising the full transporter for ~80 distinct ABC-A individuals (from S1 Fig). Sequences for nucleotide binding and inner leaflet ligand binding were recognized from the high-resolution ABCA4 structure and are nearly invariantly conserved for the entire sub-family, as indicated by red color. "Cross over" and "Cross back" sequences are also fundamentally conserved for the entire sub-family.
(DOCX)

**S3 Fig. Conservation of ligand binding in the inner leaflet of ABCA1 and ABCA4. Panel A:** partial cholesterol ligand was modeled in ABCA1 (our structure) near residues (Leu745, Phe1347, Leu1353, Phe1667, Tyr1767 and Val1768) as supported by original electron density. **Panel B:** full cholesterol molecule, as modeled by the original authors, in the corresponding site of ABCA4 (Leu760, Lys1371, Leu1379, Phe1692, Tyr1792 and Val1793). **Panel C:** partial cholesterol ligand was model in ABCA1 (our structure) near residues Thr1765, Val1769 and Ser1772 supported by original electron density. **Panel D:** partial cholesterol molecule, as modeled by the original authors, in the corresponding site of ABCA4 near residues Thr1790, Val1793 and Cys1797.
(DOCX)

**S1 Table. Secondary structure calculation of 31 different ABC transporters measured directly from the indicated PDB depositions.** Alpha-carbons with data type as helix (i.e. "H") or sheet (i.e. "S") were taken directly from their secondary structure assignment within PyMOL. The average percentage ± SD of β-sheet for all 11 eukaryotic transporters was 7.8 ± 1.5%.
(DOCX)

# Acknowledgments

**Coordinates**

The improved structures of human ABCA1 has been deposited to the Protein Data Bank with accession code 7ROQ.

**Disclosure and acknowledgments**

The authors declare no competing financial interests or conflicts of interest. We contacted the publishing authors of the 2017 ABCA1 structure and we thank them for their responses. We wish to thank Dr. Jay Heinecke for providing critical suggestions for the manuscript. During the preparation of this manuscript, we were made aware of the ABCA1 structure prediction from AlphaFold [43] that includes domain-swapped regulatory domains.

# Author Contributions

**Conceptualization:** Stephen G. Aller.

**Formal analysis:** Stephen G. Aller.

**Funding acquisition:** Stephen G. Aller, Jere P. Segrest.

**Investigation:** Stephen G. Aller.

**Methodology:** Stephen G. Aller.

**Validation:** Stephen G. Aller.

**Writing – original draft:** Stephen G. Aller.

**Writing – review & editing:** Stephen G. Aller, Jere P. Segrest.

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
