## [Decision Letter · Decision Letter 0]

15 Dec 2021

PONE-D-21-34405The Regulatory Domains of the Lipid Exporter ABCA1 Form Domain Swapped LatchesPLOS ONE

Dear Dr. Aller,

Thank you for submitting your manuscript to PLOS ONE. After careful consideration, we feel that it has merit but does not fully meet PLOS ONE’s publication criteria as it currently stands. Therefore, we invite you to submit a revised version of the manuscript that addresses the points raised during the review process.

We look forward to receiving your revised manuscript.

Kind regards,

Steffi Sun

Academic Editor

PLOS ONE

Journal Requirements:

(The authors declare no competing financial interests or conflicts of interest.  We acknowledge funding support from NIH grant P01 HL128203 to SGA and JPS. We contacted the publishing authors of the 2017 ABCA1 publication and we thank them for their responses. We wish to thank Dr. Jay Heinecke for providing critical suggestions for the manuscript. During the preparation of this manuscript, we were made aware of the ABCA1 structure prediction from AlphaFold [43] that includes domain-swapped regulatory domains.)

(Funded by NIH grant P01HL128203. The funders had no role in study design, data collection and analysis, decision to publish, or preparation of the manuscript.)

Reviewers' comments:

Reviewer's Responses to Questions

**Comments to the Author**

1. Is the manuscript technically sound, and do the data support the conclusions?

Reviewer #1: Partly

Reviewer #2: Yes

2. Has the statistical analysis been performed appropriately and rigorously? 

Reviewer #1: I Don't Know

Reviewer #2: Yes

3. Have the authors made all data underlying the findings in their manuscript fully available?

Reviewer #1: Yes

Reviewer #2: Yes

4. Is the manuscript presented in an intelligible fashion and written in standard English?

Reviewer #1: Yes

Reviewer #2: Yes

5. Review Comments to the Author

Reviewer #1: In this manuscript “The Regulatory Domains of the Lipid Exporter ABCA1 Form Domain Swapped Latches”, the authors presented a compelling case for the correction of the regulatory domains in a recently published cryo-EM model of the exporter ABCA1 (ref 25). Through sequence conservation analysis across ABCA subfamily members and re-refinement of the structural model of ABCA1 using the original cryo-EM density map published in ref 25, the authors proposed that regulatory domains of ABCA1 also form a domain-swapped latch, similar to the ones observed in the cryo-EM structures of the importer ABCA4 (ref 23, 24). Although there was no new dataset being reported, the implications of the findings are potentially important, which suggest that ABC-A subfamily members might share this domain-swap feature in their regulatory domains. Overall, the paper is well-written and the evidence is clearly presented.

I do have some comments below:

Major:

In line 179-180, “We conclude 1) ABCA1 contains domain-swapped regulatory domains that appear to function like latches, and 2) the domain swapping is conserved in the entire ABC-A subfamily.”

Although the authors have built a strong case to support the conclusion 1), it is challenging to build an accurate molecular model involving domain-swap, with the current cryo-EM map of ABCA1 at resolution worse than 4 Å (ref. 25). To be more conclusive about the configuration of the regulatory domains in ABCA1, structures with better resolution are likely necessary. Also worth noting, the structure of ABCG5/G8 (Type V) (ref. 30) does not have the domain-swapped regulatory domains.

The conclusion 2) is built upon conclusion 1), given the evidence found in sequence alignment. But if conclusion 1) is not conclusive, so is the conclusion 2).

Suggestion: soften the language in this statement. For example, instead of “We conclude: …”, use “We propose: …”. Change “2) the domain swapping is conserved …” to “2) the domain swapping is likely conserved …”

Minor:

1. In Figure 2 left panel, instead of using one black line indicating the pseudo two-fold axis of symmetry, two paralleled black line might be better suited to indicate the “non-crossing” nature of the two regulatory domains. In fact, the black lines in the right panel are used to indicate the axis of connection between the NDBs and RDs, not the axis of any symmetry.

2. In line 122-123, “Multi-sequence alignment of the RDs showed remarkable conservation of fundamental structure-function of the ABC-A subfamily, including RD crossover and bridging sequences (Fig. 4)”. The sequence alignment shows sequence conservation in “crossover” and “crossback” region, but it is not clear to me what “fundamental structure-function” means in this context. Please clarify.

3. In Figure 4 panel B, RDs should be labelled on the left side, which is at the N-termini of the crossover sequence. NBDs should be on the right side.

4. In line 123, “Several amino acid residues in that region are completely invariant (Fig 4, panel B).” Based on the re-refined model, can author comment on these invariants, such as their potential roles in stabilizing the structure of the crossover region?

5. In line 180, “We speculate that the common domain-swapped architecture may imply a mechanism of transport for ABC-A that is quite distinct from the canonical alternating access model”.

Are authors implying that the domain-swapped is incompatible with the canonical alternating access model? Is it possible the flexibility between the NBDs and RDs would still allow the NBDs to separate at certain conformational state, therefore still complying with the alternating access model? If distinct from alternating access model, what are the potential mechanisms, any hypothesis or references?

Author should elaborate on this speculation a little more. A couple of sentences will help.

Reviewer #2: The current manuscript describes the refinement and model correction of the previously published structure of the non-swapped regulatory domain of the ABC1 exporter. The study, supported by phylogenetic analyses and insights from the Alpha fold database, reports that the regulatory domain should be in swapped arrangements. This is a revised manuscript, and the authors have answered the comments and concerns raised by reviewers. Overall the manuscript is very well written, and the analysis is thoughtful and convincing. I don’t have further comments except for two minor typological errors given below. I recommend that now the manuscript can be formally accepted.

Minor comments.

• Page 4, line 98: There is a small typo error, “lathes” should be written as “latches”

Minor comments: At some places, authors wrote “subfamilies” and “sub-families.” It would be better to keep it consistent throughout the manuscript.

6. PLOS authors have the option to publish the peer review history of their article (what does this mean?). If published, this will include your full peer review and any attached files.

Reviewer #1: No

Reviewer #2: No

---

## [Author Response · Author response to Decision Letter 0]

16 Dec 2021

PONE-D-21-34405

Aller and Segrest

“The Regulatory Domains of the Lipid Exporter ABCA1 Form Domain Swapped Latches”

Comments in black font were copied verbatim from the reviewer’s text.

Author responses are colored blue below.

“The current manuscript describes the refinement and model correction of the previously published structure of the non-swapped regulatory domain of the ABC1 exporter. The study, supported by phylogenetic analyses and insights from the Alpha fold database, reports that the regulatory domain should be in swapped arrangements. This is a revised manuscript, and the authors have answered the comments and concerns raised by reviewers. Overall the manuscript is very well written, and the analysis is thoughtful and convincing. I don’t have further comments except for two minor typological errors given below. I recommend that now the manuscript can be formally accepted.”

We thank the reviewer for their time and effort in providing an expert critique of our manuscript, and particularly for the comment “I recommend that now the manuscript can be formally accepted.”

Minor comments.

• Page 4, line 98: There is a small typo error, “lathes” should be written as “latches”

o We have made this correction (line 98).

• Minor comments: At some places, authors wrote “subfamilies” and “sub-families.” It would be better to keep it consistent throughout the manuscript.

o We thank the Reviewer for this attention to detail. We had written “subfamily/subfamilies” 12 times and “sub-family/sub-families” 6 times in our manuscript. To be consistent, we have changed all instances of “sub-family/sub-families” to “subfamily/subfamilies”, i.e. removing the dash (lines 135, 138, 152, 183, 392, and 431).

---

## [Editor Report · Decision Letter 1]

5 Jan 2022

The Regulatory Domains of the Lipid Exporter ABCA1 Form Domain Swapped Latches

PONE-D-21-34405R1

Dear Dr. Aller,

We’re pleased to inform you that your manuscript has been judged scientifically suitable for publication and will be formally accepted for publication once it meets all outstanding technical requirements.

Kind regards,

Steffi Sun

Academic Editor

PLOS ONE
---

## [Editor Report · Acceptance letter]

27 Jan 2022

PONE-D-21-34405R1 

The Regulatory Domains of the Lipid Exporter ABCA1 Form Domain Swapped Latches 

Dear Dr. Aller:

I'm pleased to inform you that your manuscript has been deemed suitable for publication in PLOS ONE. Congratulations! Your manuscript is now with our production department. 

Kind regards, 

on behalf of

Dr. Qiu Sun 

Academic Editor

PLOS ONE